# Acute kidney injury in COVID-19 pediatric patients in North America: Analysis of the virtual pediatric systems data

**Rupesh Raina**[1,2]\*, **Isabelle Mawby**[3], **Ronith Chakraborty**[1,2], **Sidharth Kumar Sethi**[4], **Kashin Mathur**[5], **Shefali Mahesh**[1], **Michael Forbes**[6]

**1** Department of Nephrology, Akron Children's Hospital, Akron, OH, United States of America, **2** Department of Nephrology, Akron Nephrology Associates/ Cleveland Clinic Akron General Medical Center, Akron, OH, United States of America, **3** Department of Medicine, Northeast Ohio Medical University, Rootstown, OH, United States of America, **4** Pediatric Nephrology & Pediatric Kidney Transplantation, Kidney and Urology Institute, Medanta, The Medicity Hospital, Gurgaon, India, **5** Research Student, Akron Nephrology Associates/ Cleveland Clinic Akron General Medical Center, Akron, OH, United States of America, **6** Department of Pediatrics, Akron Children's Hospital, Akron, OH, United States of America

\* rraina@akronchildrens.org, raina@akronnephrology.com

**Data Availability Statement:** All relevant data are within the manuscript and its Supporting Information files.

## Abstract

### Background

Despite extensive research into acute kidney injury (AKI) in adults, research into the epidemiology, associated risk factors, treatment, and mortality of AKI in pediatric COVID-19 patients is understudied. Advancing understanding of this disease is crucial to further developing treatment and preventative care strategies to reduce morbidity and mortality.

### Methods

This is a retrospective analysis of 2,546 COVID-19 pediatric patients (age $\leq$ 21 years) who were admitted the ICU in North America. Analysis of the Virtual Pediatric Systems (VPS) COVID-19 database was conducted between January 1, 2020, and June 30, 2021.

### Results

Out of a total of 2,546 COVID positive pediatric patients, 10.8% (n = 274) were diagnosed with AKI. Significantly higher continuous and categorical outcomes in the AKI subset compared to the non-AKI cohort included: length of stay at the hospital (LOS) [9.04 (5.11–16.66) vs. 5.09 (2.58–9.94) days], Pediatric Index of Mortality (PIM) 2 probability of death [1.20 (0.86–3.83) vs. 0.96 (0.79–1.72)], PIM 3 probability of death [0.98 (0.72–2.93) vs. 0.78 (0.69–1.26)], mortality [crude OR (95% CI): 5.01 (2.89–8.70)], airway and respiratory support [1.63 (1.27–2.10)], cardio-respiratory support [3.57 (1.55–8.23)], kidney support [12.52 (5.30–29.58)], and vascular access [4.84 (3.70–6.32)].

### Conclusions

This is one of the first large scale studies to analyze AKI among pediatric COVID-19 patients admitted to the ICU in North America. Although the course of the COVID-19 virus appears

**Funding:** The authors received no specific funding work.

**Competing interests:** The authors have declared that no competing interests exist.

milder in the pediatric population, renal complications may result, increasing the risk of disease complication and mortality.

## Introduction

Severe acute respiratory syndrome coronavirus 2 (SARS-CoV-2) is responsible for the 2019 coronavirus pandemic (COVID-19). According to The World Health Organization, over 182,319,261 cases and almost 4 million deaths have been reported worldwide as of July 2021 [1]. Kidney manifestations from COVID-19 are becoming increasingly prevalent, with acute kidney injury (AKI) contributing to high mortality rates. [2, 3].

Previously, we performed a meta-analysis assessing AKI incidence and outcomes among both pediatric and adult COVID-19 patients. Among 1,247 COVID-19 positive (+) pediatric patients, 30.51% developed AKI. Among the COVID-19+ patients with AKI, 0.56% received kidney replacement therapy (KRT). In comparison, among the studied adult COVID-19 population (n = 42,591), 15.90% developed AKI. The overall mortality rates in children and adults were 2.55% and 14.60%, respectively [4]. However, despite this age-related discrepancy, COVID-19's renal association has been a consistent, negative, prognostic factor [4–7].

Throughout the world, preliminary studies are starting to assess the epidemiology of AKI in pediatric population affected by COVID-19, with reported incidents ranging from 0.8% to 80% [8–21]. However, there are no large-scale studies examining this association in North America. This study uses the Virtual Pediatric Systems database (VPS) to identify the epidemiology, associated risk factors, treatments, and mortality of AKI in pediatric COVID-19 patients admitted to the pediatric intensive care unit (PICU) in North America.

## Methods

### Study design and population

A retrospective analysis of 2,597 COVID-19 pediatric patients (age ≤ 21 years) in the ICU within North America was performed by analyzing the VPS COVID-19 database, which includes data from over 200 pediatric critical care hospital units, between January 1, 2020 and June 30, 2021. Male and female patients <21 years old with AKI were analyzed (determined by Kidney Disease Improving Global Outcomes (KDIGO) staging of stage 1 or worse), along with patients with current or recent SARS-CoV-2 infection (determined by RT-PCR, serology, or antigen test), and ICU admission. Excluded from the study were patients >21 years old and patients with a negative SARS-CoV-2 status.

### Data collection

Data regarding pediatric patients < 21 years old with a primary diagnosis of COVID-19 and AKI diagnosis was recorded. Patients with COVID-19 were identified using ICD-10, which also identified any presenting underlying conditions. Patients negative for COVID-19 were excluded from the study. Variables utilized in the study included COVID-19 positivity, confirmed deaths, PICU days, therapies used, comorbidities of patients, cumulative COVID-19 positive PICU admission, COVID-19 positive PICU admissions per day, average LOS by age group, average LOS by period comorbidities, and organ support. S1B Table in S1 File provides the definition of these variables. In order to investigate epidemiology, risk factors associated with AKI, results on treatment, and associated mortality in the pediatric population, the

following variables were necessary: patient demographics (age, sex, race, ethnicity (White, Black, Hispanic, Asian/Pacific Islander, etc.), diagnosis (primary and secondary), lab order/ results (hematology and chemistry tests), image order/results, respiratory support modality, kidney replacement therapy modality (if utilized), medications, interventions and procedures, pediatric index of mortality (PIM) 2 and 3, discharge status, length of stay, and mortality. PIM 2 and 3 are severity scoring systems consisting of 10 variables which are used for predicting outcome of patients admitted to intensive care units (ICUs). It provides predicted % of mortality rate, where higher indicates greater probability of death [22]. The patients were separated into an AKI group and a non-AKI group. In the AKI group, patients were subcategorized based on staging of AKI severity by KDIGO guidelines. PIM3, KRT, and mortality data was used when urine output and serum creatinine level data were not available. If KRT was utilized, the patient was graded as AKI stage 2. Patients with the presence of mortality status were graded as AKI stage 3. The baseline parameters among AKI and non-AKI patients are provided in Table 1. The Pediatric Overall Performance Category (POPC) and the Pediatric Cerebral Performance Category (PCPC) scales are included as baseline parameters. Both the POPC and the PCP are qualitative assessments of performance, conducted by observers, based on the Glasgow Outcome Scale. Scores include 1 for good, 2 for mild disability, 3 for moderate disability, 4 for severe disability, and 5 for vegetative state or coma (6 indicates death, but was not included in the study) [23, 24].

The laboratory profile of patients among the AKI and non-AKI group are included in Table 2. EGFR was calculated using IDMS-Traceable Schwartz Equation [eGFR = (0.41 x height in cm) ÷ serum Cr]. Systolic and Diastolic blood pressure Z score was calculated using the equation provided in National High Blood Pressure Education Program Working Group on High Blood Pressure in Children and Adolescents [25]. In Table 1, BMI z score was calculated based upon the CDC's definition. Obesity is defined as BMI z score > 1.64 [18]. The association of different outcomes among AKI patients across different AKI stages are included in Table 4. The continuous outcome variables include hospital length of stay, PIM 2 probability of death and PIM 3 probability of death. Categorical outcomes include mortality, airway and respiratory support, cardiorespiratory support, kidney support, and vascular access. Vascular access included arterial catheter; Hemodialysis/ Plasmapheresis catheters; ECMO cannula; venous catheter (peripherally inserted central catheter; and percutaneous central venous catheter). Airway/respiratory support included endotracheal intubation (and duration of intubation); tracheostomy tube insertion (includes duration of intubation); invasive ventilation [mechanical ventilation, conventional (including CPAP plus pressure support); CPAP (invasive); HFOV (high frequency oscillator ventilation); Jet Ventilation]; Non-Invasive Respiratory Support or Ventilation [BiPAP (non-invasive); Mechanical Ventilation (Non-Invasive); CPAP (Non-Invasive); and Humidified High Flow Oxygen. Cardio-respiratory support included ECMO. Kidney support included Intermittent Hemodialysis (IHD).

This study has been approved by the Akron Children's Health Centers IRB Committee. The project approval number is 1632930-AKHC. This is a retrospective study of medical records, and the IRB committee waived the requirement for informed consent. All data was fully anonymized before being accessed.

## Statistical analysis

All the variables were tested for normality using Kolmogorov-Smirnov test. Categorical variables were summarized as frequencies and percentages, while continuous variables were summarized as medians and inter-quartile range (IQR; 25th to 75th percentiles). Univariate analysis (using the chi-square or Fischer exact test for categorical variables and Mann Whitney

**Table 1. Baseline parameters among AKI and non-AKI group.**

| Variables | | AKI | Non-AKI | p value |
|---|---|---|---|---|
| | | N (%) | N (%) | |
| Age | Neonate Birth to 29 days | 2 (0.7%) | 44 (1.9%) | <0.001 |
| | Infant 29 days to < 2 years | 18 (6.6%) | 340 (15.0%) | |
| | Child 2 to < 6 years | 21 (7.7%) | 319 (14.0%) | |
| | Child 6 to < 12 years | 59 (21.5%) | 476 (21.0%) | |
| | Adolescent 12 to < 18 years | 152 (55.5%) | 943 (41.5%) | |
| | Adolescent (late) 18 to < 21 years | 22 (8.0%) | 150 (6.6%) | |
| Age in median (IQR) [years] | | 5 (4–5) | 4 (3–5) | <0.001 |
| Gender | Male [vs. female] | 142 (51.8%) | 1248 (54.9%) | 0.33 |
| Race | White [vs. non-white] | 59 (25.2%) | 543 (27.6%) | 0.433 |
| Ethnicity | Hispanic / Latino [vs. non Hispanic / Latino] | 88 (37.6%) | 746 (38%) | 0.915 |
| Readmission patient | Yes [vs. No] | 70 (25.5%) | 612 (26.9%) | 0.624 |
| Co-morbidities | Cardiovascular [vs. No] | 161 (58.8%) | 721 (31.7%) | <0.001 |
| | Endocrine [vs. No] | 81 (29.6%) | 389 (17.1%) | <0.001 |
| | Gastrointestinal [vs. No] | 64 (23.4%) | 428 (18.8%) | 0.073 |
| | Haematology [vs. No] | 124 (45.3%) | 514 (22.6%) | <0.001 |
| | Neurologic [vs. No] | 87 (31.8%) | 761 (33.5%) | 0.563 |
| | Oncology [vs. No] | 8 (6.8%) | 50 (5.3%) | 0.495 |
| | Respiratory [vs. No] | 176 (64.2%) | 1257 (55.3%) | 0.005 |
| | Obesity [vs. No] | 79 (54.1%) | 653 (58.1%) | 0.353 |
| Baseline Paediatric Cerebral Performance Category | Normal | 27 (84.4%) | 287 (70.3%) | 0.527 |
| | Mild disability | 2 (6.2%) | 46 (11.3%) | |
| | Moderate disability | 2 (6.2%) | 33 (8.1%) | |
| | Severe disability | 1 (3.1%) | 41 (10%) | |
| | Coma or vegetative state | 0 (0%) | 1 (0.2%) | |
| Baseline Paediatric Overall Performance Category | Normal | 23 (71.9%) | 240 (58.8%) | 0.491 |
| | Mild disability | 5 (15.6%) | 63 (15.4%) | |
| | Moderate disability | 3 (9.4%) | 62 (15.2%) | |
| | Severe disability | 1 (3.1%) | 42 (10.3%) | |
| | Coma or vegetative state | 0 (0%) | 1 (0.2%) | |

AKI: Acute Kidney Injury

U test for continuous variables) was carried out to assess the unadjusted relationship between the variables / different outcomes in the two groups. The multivariate linear or logistic regression was conducted to assess the association of different outcomes in the two groups after adjusting for variables observed to be significant in the univariate analysis. The difference in outcomes across three AKI staging group was assessed using Kruskal Wallis test for continuous variables and chi-square or Fischer exact test for categorical variables. A two-sided p value <0.05 was statistically significant. Statistical software (SPSS version 20) was used to perform the statistical analyses.

## Results

A total of 2,546 COVID+ patients were included in our analysis where 10.8% (n = 274) had a diagnosis of AKI. In this subset, 62.8% (n = 172) had stage 1 AKI, 14.6% (n = 40) had stage 2 AKI, and 22.6% (n = 62) had stage 3 AKI. The median (IQR) age (years) of the patients with AKI was observed to be significantly higher versus without AKI [5 (4–5) vs. 4 (3–5)].

**Table 2. Laboratory profile of patients among AKI and non-AKI group.**

| Variables | AKI | | Non-AKI | | p value |
|---|---|---|---|---|---|
| | N | Median (IQR) | N | Median (IQR) | |
| BMI z score | 146 | 1.9 (0.4–3.3) | 1123 | 2.2 (0.4–3.7) | 0.425 |
| SBP (mmHg) | 118 | 118.5 (106–133.3) | 941 | 117 (105–128.5) | 0.267 |
| SBP z score^ | 64 | 2 (0.9–3.1) | 441 | 1.9 (0.9–3.1) | 0.986 |
| DBP (mmHg) | 118 | 71 (61.8–85) | 942 | 72 (62–82) | 0.929 |
| DBP z score^ | 64 | 1.5 (0.5–2.8) | 441 | 1.7 (0.9–2.8) | 0.109 |
| Heart Rate (bpm) | 118 | 128 (114.8–148) | 947 | 127 (107–147) | 0.107 |
| Respiratory Rate (bpm) | 118 | 36.5 (26.8–47) | 939 | 34 (26–44) | 0.092 |
| Temperature (˚C) | 118 | 37.2 (36.8–38.5) | 932 | 37.2 (36.8–37.8) | 0.361 |
| pH | 76 | 7.3 (7.2–7.4) | 345 | 7.4 (7.3–7.4) | <0.001 |
| pCO2 (mmHg) | 78 | 37.3 (30.4–46.2) | 336 | 39.5 (33.6–47) | 0.079 |
| Haemoglobin (g/dL) | 71 | 12.1 (10.4–14.3) | 399 | 11.9 (10.5–13.5) | 0.411 |
| WBC ($10^9$/L) | 66 | 15.9 (9–23.5) | 334 | 9.4 (6.3–13.6) | <0.001 |
| Platelet Count ($10^9$/L) | 58 | 168 (110.3–273.5) | 326 | 216.5 (149–289.5) | 0.07 |
| PT (Seconds) | 49 | 15.2 (13.4–17.4) | 237 | 14.5 (13.1–16.2) | 0.278 |
| PTT (Seconds) | 45 | 30.9 (27.1–44.9) | 226 | 32 (28.8–37) | 0.919 |
| Sodium (Serum) (mmol/L) | 99 | 143 (137–148) | 495 | 141 (137–145) | 0.064 |
| Potassium (Serum) (mmol/L) | 100 | 4.6 (3.9–5.7) | 481 | 4.3 (3.8–4.9) | 0.008 |
| Bicarbonate (mmol/L) | 77 | 19 (14–22.3) | 430 | 21.1 (18–24.2) | <0.001 |
| Blood Urea Nitrogen (mmol/L) | 91 | 9.3 (5.4–15.7) | 452 | 3.9 (2.9–5.4) | <0.001 |
| Creatinine (mg/dL) | 91 | 1.3 (0.8–2.4) | 451 | 0.5 (0.3–0.7) | <0.001 |
| eGFR (ml/min/1.73m²)* | 51 | 41.7 (23.7–68.2) | 218 | 107.6 (82–146.3) | <0.001 |
| Glucose (Serum) (mg/dL) | 100 | 157 (125–260) | 505 | 128 (106–192) | <0.001 |
| Total Calcium (mg/dL) | 75 | 8.4 (7.8–9.4) | 404 | 8.6 (8.1–9.2) | 0.325 |
| Total Bilirubin (mg/dL) | 57 | 0.5 (0.3–1.3) | 296 | 0.5 (0.3–0.8) | 0.322 |
| Albumin (g/dL) | 53 | 3 (2.6–3.8) | 310 | 3.5 (2.8–4) | 0.062 |

AKI: Acute Kidney Injury; mmHg: millimeter of mercury; bpm: beat per minute; ˚C: degree Celsius; pCO2: partial pressure of carbon dioxide; g: gram; dl: deciliter; WBC: White blood cells; L: liter; PT: prothrombin time; PTT: partial thromboplastin time; mmol/L: millimoles per liter; mg: milligram; IQR: Interquartile range

^calculated only for children with height data availability

*calculated only for children with height and creatinine data availability

Associated co-morbidities with AKI—according to the VPS database–include respiratory [64.2% vs. 55.3%; p = 0.005], cardiovascular [58.8% vs. 31.7%; p<0.001], endocrinal [29.6% vs. 17.1%; p<0.001], and hematology [45.3% vs. 22.6%; p<0.001] dysfunctions when compared with non-AKI cohort. Notably, the gender, race, ethnicity, baseline pediatric cerebral performance category, and baseline pediatric overall performance category were not statistically significant between the two groups (Table 1).

Clinical lab values which were higher in the AKI subset compared to the non-AKI cohort are reported as 'median (IQR)' and include: white blood cells count, serum potassium, blood urea nitrogen, creatinine, eGFR, and serum glucose. pH and bicarbonate levels were significantly lower in the AKI subset compared to the non-AKI cohort (Table 2).

Significantly higher continuous and categorical outcomes in the AKI subset compared to the non-AKI cohort included: length of stay at the hospital (LOS) [9.04 (5.11–16.66) vs. 5.09 (2.58–9.94) days; p<0.001], PIM 2 probability of death [1.20 (0.86–3.83) vs. 0.96 (0.79–1.72)%; p<0.001], PIM 3 probability of death [0.98 (0.72–2.93) vs. 0.78 (0.69–1.26)%; p<0.001], mortality [crude OR (95% CI): 5.01 (2.89–8.70)], airway and respiratory support [1.63 (1.27–

2.10)], cardio-respiratory support [3.57 (1.55–8.23)], kidney support [12.52 (5.30–29.58)], and vascular access [4.84 (3.70–6.32)] (S1A and S1B Figs in S1 File).

The continuous and categorical outcomes remained greater in the AKI versus non-AKI cohort even after adjusting for variables significant in the uni-variate analysis (such as age (continuous), presence of cardiovascular, endocrine, hematology, and respiratory co-morbidities) and included: LOS [6.29 (3.95–8.64) days], the adjusted odds (95% CI) of mortality [2.69 (1.48–4.88)], of airway and respiratory support [1.61 (1.16–2.24)], of kidney support [5.34 (2.15–13.25)], and of vascular access [3.51 (2.63–4.70)] (Table 3). The adjusted association of other variables with the outcomes is shown in S1C Table in S1 File.

When comparing the continuous outcomes to the stages of AKI, the values increased statistically significantly for LOS [stage 1: 8.24 (4.90–13.6); stage 2: 8.78 (4.64–16.05); and stage 3: 11.93 (7.40–24.33) days; p = 0.009], but not for PIM 2 [1.15 (0.89–3.36) vs. 1.22 (0.82–5.24) vs. 1.28 (0.85–4.42)%; p = 0.606], and PIM 3 [0.96 (0.73–2.34) vs. 1.04 (0.38–2.97) vs. 1.12 (0.7–4.7)%; p = 0.484]. However, when comparing the categorical outcomes to the stages of AKI, airway and respiratory support reported statistically significant data [stage 1: 50.0%; stage 2: 55.0%; and stage 3: 71.0%; p = 0.019] and need of vascular access [59.3% vs. 70.0% vs. 87.1%; p<0.001]. The mortality status and kidney support has been used in the classification of AKI stage; therefore these outcomes have not been considered for analysis (Table 4) A multivariate analysis for these outcomes based on AKI staging was not conducted due to the limited sample size.

## Discussion

As of July 2021, the public, online VPS dashboard reported 2,596 pediatric patients with COVID-19 admitted to North American PICUs, of which approximately 3.04% required KRT. Within this subset, 86.4% were successfully discharged afterwards but 13.6% eventually died. From the patients that received KRT, 84.8% reported respiratory dysfunction, 81.2% reported circulatory dysfunction, and 42.4% reported vascular dysfunction. Notably, 14.0% of the total cohort reported kidney/urinary organ dysfunction, of which 88.8% were discharged and 11.2% died.

Among 2,546 COVID-19+ children, 10.8% of patients developed AKI. This coincides with the incidence reported by Derespina *et al.* in a study at New York City PICUs; Among 70 children admitted, 12.9% developed AKI [21]. Separate studies from Saudi Arabia, Iran, England, and France found much higher incidences of AKI at 21%, 22%, 29%, 19%, and 70% respectively [13, 15–17, 26] (**S1A Table in S1 File**). Contrarily, studies from Italy, Spain, and two from China all reported AKI incidences significantly lower at 1.2%, 0.8%, 1.3%, and 2.7%, respectively [8, 9, 11, 27] (**S1A Table in S1 File**). A recent study by Basu et al. reported an AKI incidence of 37.5% among 311 pediatric patients with COVID-19 [28]. However, only 51.4% of the patients had a confirmed SARS-CoV-2 infection, while the rest of the population were suspected cases. The authors acknowledge the presence of a possible selection bias, as provides may have been more likely to suspect SARS-CoV-2 with a negative test result in more severely ill patients than those who were less severely ill. Kari et al. hypothesized that some discrepancy between AKI incidences may also be attributed to differences in the AKI definitions used [13]. Some studies used the KDIGO definition to define AKI; however, an English study by Stewart et al. analyzed the British Association of Pediatric Nephrology's (BAPN) and found a higher incidence of 29% [26].

The VPS data showed that patients in the AKI group have a much higher rate of mortality [crude OR (95% CI): 5.01 (2.89–8.70)]. Likewise, a study from Saudi Arabia found a mortality rate of 42% among AKI patients and 0% among non-AKI patients [13]. The VPS results

**Table 3. Univariate (crude) and adjusted association of different outcomes among patients with versus without AKI.**

| Univariate Association | | | | | |
|---|---|---|---|---|---|
| **Continuous outcome** | | | | | |
| **Outcomes** | **AKI** | | **Non-AKI** | | **p value** |
| | **N** | **Median (IQR)** | **N** | **Median (IQR)** | |
| Hospital LOS (days) | 270 | 9.04 (5.11–16.66) | 2,228 | 5.09 (2.58–9.94) | <0.001 |
| PIM 2 Probability of Death (%) | 274 | 1.20 (0.86–3.83) | 2,272 | 0.96 (0.79–1.72) | <0.001 |
| PIM 3 Probability of Death (%) | 274 | 0.98 (0.72–2.93) | 2,272 | 0.78 (0.69–1.26) | <0.001 |
| **Categorical outcome** | | | | | |
| **Outcomes** | **AKI [N (%)]** | **Non-AKI [N (%)]** | **p value** | **Crude Odds ratio (95% CI)** | |
| Mortality | 21 (7.7%) | 37 (1.6%) | <0.001 | 5.01 (2.89–8.70) | |
| Airway / Respiratory support | 152 (55.5%) | 983 (43.3%) | <0.001 | 1.63 (1.27–2.10) | |
| Cardio-respiratory support | 8 (2.9%) | 19 (0.8%) | 0.006 | 3.57 (1.55–8.23) | |
| Kidney support | 13 (4.7%) | 9 (0.4%) | <0.001 | 12.52 (5.30–29.58) | |
| Vascular access | 184 (67.2%) | 675 (29.7%) | <0.001 | 4.84 (3.70–6.32) | |
| Adjusted Association* | | | | | |
| **Continuous outcome** | | | | | |
| **Outcomes** | **Un-standardized Coefficients (95% CI)^** | | | **p value** | |
| Hospital LOS (days) | 6.29 (3.95–8.64) | | | <0.001 | |
| PIM 2 Probability of Death (%) | 0.81 (-0.11–1.72) | | | 0.083 | |
| PIM 3 Probability of Death (%) | 0.61 (-0.30–1.51) | | | 0.187 | |
| **Categorical outcome** | | | | | |
| **Outcomes** | **Adjusted odds ratio (95% CI)^** | | | **p value** | |
| Mortality | 2.69 (1.48–4.88) | | | 0.001 | |
| Airway / Respiratory support | 1.61 (1.16–2.24) | | | 0.005 | |
| Cardio-respiratory support | 1.88 (0.77–4.56) | | | 0.165 | |
| Kidney support | 5.34 (2.15–13.25) | | | <0.001 | |
| Vascular access | 3.51 (2.63–4.70) | | | <0.001 | |

AKI: Acute Kidney Injury; LOS: Length of stay; PIM: Pediatric index of mortality; IQR: Interquartile range; CI: Confidence Interval

^AKI versus non-AKI

*Adjusted for variables significant in the uni-variate analysis "Table 1" such as age (continuous in years), presence of cardiovascular, endocrine, hematology, and respiratory co-morbidities (categorical). The clinical and laboratory variables significant in the uni-variate analysis "Table 2" were not considered because for those variables the data was available for <25% of the children [which was leading to decrease in the sample size for adjusted analysis to a greater extent]

deviate from Bjornstad *et al*'s multicenter study throughout the United States, Eastern Europe, and Russia, where a mortality rate of 6% was found among AKI COVID-19 pediatric patients and 5% among pediatric COVID-19 patients without AKI. However, data may be underestimated as the authors note that they used creatinine instead of urine output to define AKI [12].

**Table 4. Association of different outcomes among AKI patients across different AKI stage.**

| Continuous outcome | | | | | | | |
|---|---|---|---|---|---|---|---|
| Outcomes | AKI stage 1 | | AKI stage 2 | | AKI stage 3 | | p value |
| | N | Median (IQR) | N | Median (IQR) | N | Median (IQR) | |
| Hospital LOS (days) | 168 | 8.24 (4.90–13.6) | 40 | 8.78 (4.64–16.05) | 62 | 11.93 (7.40–24.33) | 0.009 |
| PIM 2 Probability of Death (%) | 172 | 1.15 (0.89–3.36) | 40 | 1.22 (0.82–5.24) | 62 | 1.28 (0.85–4.42) | 0.606 |
| PIM 3 Probability of Death (%) | 172 | 0.96 (0.73–2.34) | 40 | 1.04 (0.38–2.97) | 62 | 1.12 (0.7–4.7) | 0.484 |
| Categorical outcome | | | | | | | |
| Outcomes | N (%) | | N (%) | | N (%) | | p value |
| Airway / Respiratory support | 86 (50.0%) | | 22 (55.0%) | | 44 (71.0%) | | 0.019 |
| Cardio-respiratory support | 5 (2.9%) | | 0 (0%) | | 3 (4.8%) | | 0.375 |
| Vascular access | 102 (59.3%) | | 28 (70.0%) | | 54 (87.1%) | | <0.001 |

AKI: Acute Kidney Injury; LOS: Length of stay; PIM: Pediatric index of mortality; IQR: Interquartile range

The mortality status and kidney support has been used in the classification of AKI stage, therefore these outcomes have not been considered for analysis.

The most common clinical features of COVID-19 among children include fever, dry cough, and pneumonia, along with an increasing prevalence of multisystem dysfunction [26]. Within the 2,596 pediatric COVID-19 cohort from the VPS data from North American PICU's, several additional organ systems were reportedly involved including 49.3% of patients with respiratory symptoms, 33.1% with circulatory symptoms, 21.6% with digestive/excretory symptom, 17.6% with hematologic symptoms, 12.3% with neurologic symptoms, and 11.7% with kidney/urinary symptom. Correspondingly, comorbidities that were reported to be significantly higher among the VPS AKI-group as opposed to the non-AKI group included respiratory [64.2% vs. 55.3%], cardiovascular [58.8% vs. 31.7%], endocrinal [29.6% vs. 17.1%], and hematologic [45.3% vs. 22.6%]. In many COVID-19 cases, residual renal impairment may be attributed to the systemic effects of decreased tissue perfusion from hypoxia or transient periods of hypotension, fluctuations in electrolyte levels or hypernatremia, and/or associated comorbidities [13]. Consequently, physicians ought to be aware of such factors that may precipitate renal complications in COVID-19 patients. Ultimately, research aims to improve short-term outcomes as measured by hospital LOS along with overall long-term outcomes in quality of life.

Despite COVID-19's potential severity, studies have suggested that children are less susceptible than adults. Lee et al. theorizes that this is because children have a more active innate immune response, are generally more protected by parents, engage in fewer outdoor activities, are less likely to travel internationally, and have relatively less comorbidities [29]. Biochemically, ACE2 demographics differ from its distribution, maturation, and function–ultimately affecting its association with the receptor binding domain of the COVID-19 spike-protein and reducing susceptibility and severity [6, 29, 30]. However, more studies are needed to fully elucidate why children are less susceptible than the adult population to severe COVID-19 infections.

Due to the retrospective database design of this study, there are notable limitations. This study is limited to data from a select number of pediatric ICUs across only North America, therefore limiting our ability to apply our results globally. Additionally, as COVID-19 continues to progress and new viral variants being to emerge, new advances in data and research will emerge, which will allow for better understanding of the virus.

## Conclusion

Overall, AKI incidence among pediatric ICU patients was 10.8%, which is higher than multiple previous large-scale studies had described. AKI severity among children with COVID-19 has

additionally been shown to be associated with an increased risk of mortality. Despite the reduced susceptibility of severe disease in children, it is crucial to continue to develop the knowledge base surrounding all manifestations of COVID-19 among children across the world. A greater understanding of COVID-19 among the pediatric population may reduce the potential for exponentially increasing rates of morbidity and mortality through this highly transmissible virus.

## Supporting information

**S1 File. Supporting information.**
(DOCX)

**S2 File. Data set.**
(XLSX)

## Acknowledgments

We would like to acknowledge Nina Vijayvargiya for her help with manuscript revision.

## Author Contributions

**Conceptualization:** Rupesh Raina, Ronith Chakraborty, Michael Forbes.

**Data curation:** Rupesh Raina, Sidharth Kumar Sethi.

**Formal analysis:** Rupesh Raina, Sidharth Kumar Sethi.

**Funding acquisition:** Rupesh Raina.

**Investigation:** Rupesh Raina.

**Methodology:** Rupesh Raina, Ronith Chakraborty, Sidharth Kumar Sethi.

**Project administration:** Rupesh Raina, Sidharth Kumar Sethi.

**Resources:** Rupesh Raina, Sidharth Kumar Sethi.

**Software:** Rupesh Raina, Sidharth Kumar Sethi.

**Supervision:** Rupesh Raina.

**Validation:** Rupesh Raina.

**Visualization:** Rupesh Raina, Ronith Chakraborty.

**Writing – original draft:** Rupesh Raina, Isabelle Mawby, Ronith Chakraborty.

**Writing – review & editing:** Rupesh Raina, Isabelle Mawby, Ronith Chakraborty, Kashin Mathur, Shefali Mahesh, Michael Forbes.

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
