## [Decision Letter · Decision Letter 0]

17 Jan 2022

PONE-D-21-39403Acute Kidney Injury in COVID-19 Pediatric Patients in North America: Analysis of the Virtual Pediatric Systems DataPLOS ONE

Dear Dr. raina,

Thank you for submitting your manuscript to PLOS ONE. After careful consideration, we feel that it has merit but does not fully meet PLOS ONE’s publication criteria as it currently stands. Therefore, we invite you to submit a revised version of the manuscript that addresses the points raised during the review process.

We look forward to receiving your revised manuscript.

Kind regards,

Zhanjun Jia

Academic Editor

PLOS ONE

Journal Requirements:

Reviewers' comments:

Reviewer's Responses to Questions

**Comments to the Author**

1. Is the manuscript technically sound, and do the data support the conclusions?

Reviewer #1: Yes

Reviewer #2: Yes

2. Has the statistical analysis been performed appropriately and rigorously? 

Reviewer #1: Yes

Reviewer #2: I Don't Know

3. Have the authors made all data underlying the findings in their manuscript fully available?

Reviewer #1: Yes

Reviewer #2: Yes

4. Is the manuscript presented in an intelligible fashion and written in standard English?

Reviewer #1: Yes

Reviewer #2: Yes

5. Review Comments to the Author

Reviewer #1: Dear Authors,

I have reviewed your manuscript titled "Acute Kidney Injury in COVID-19 Pediatric Patients in North America: Analysis of the Virtual Pediatric Systems Data" with keen interest.

I have very few comments.

1. It is usually best to leave continuous variables uncategorized, except where such categorization are standard e.g categorizing blood sugar levels according to diabetes defining cut-off values. It seems arbitrary that in some of the analyses, age is categorized as <12 years or >12 years (line 116) while in other instances it is categorized as <2 years or >2 years (line 146 and Table 1). The <2 years or >2 years categorization lumps up individuals with different anatomic/developmental and physiologic profiles. The group >2 years would comprise for instance 3-5 year olds, 6-10 year olds, young adolescents and late adolescents, given the cut off age in the study of <21 years.

If the age is left as a continuous variable, you could then assess what the odds ratios would be for different ages with incremental age and this would provide more clinically applicable and relevant deductions.Kindly have a look at this.

2. In line 168, you mention "circulatory dysfunction" and "vascular dysfunction". In reading through the manuscript I did not come across where these entities had been clearly defined and distinguished.

Overall, it is a nicely written, straight-to-the-point manuscript.

Thank you.

Reviewer #2: Thank you for your submission. I am curious to know why pediatric age rage was unto 21-years? It may be important to know that the studies that you have refereed/compared to have used the similar upper age. In line 15 PIM abbreviation is used for the first time. without its full name kindly correct.

6. PLOS authors have the option to publish the peer review history of their article (what does this mean?). If published, this will include your full peer review and any attached files.

Reviewer #1: No

Reviewer #2: **Yes: **Aasim Ahmad

---

## [Author Response · Author response to Decision Letter 0]

8 Feb 2022

Review’s Comments:

Reviewer 1

Reviewer Comment: It is usually best to leave continuous variables uncategorized, except where such categorization are standard e.g categorizing blood sugar levels according to diabetes defining cut-off values. It seems arbitrary that in some of the analyses, age is categorized as <12 years or >12 years (line 116) while in other instances it is categorized as <2 years or >2 years (line 146 and Table 1). The <2 years or >2 years categorization lumps up individuals with different anatomic/developmental and physiologic profiles. The group >2 years would comprise for instance 3-5 year olds, 6-10 year olds, young adolescents and late adolescents, given the cut off age in the study of <21 years. If the age is left as a continuous variable, you could then assess what the odds ratios would be for different ages with incremental age and this would provide more clinically applicable and relevant deductions. Kindly have a look at this.

Response: We thank the reviewer very much for their suggestion. We have re-done our data analysis as suggested so that age may be analyzed as a continuous variable with odds ratios. Kindly view the changes in the statistical analysis portion of the methods section, the results section, Table 1, and Table 3 of the manuscript. We have included S3 Table under supporting information entitled “Adjusted association of different variables on the outcomes” so that the adjusted association of age as a continuous variable with the different outcomes is shown.

Reviewer Comment: In line 168, you mention "circulatory dysfunction" and "vascular dysfunction". In reading through the manuscript I did not come across where these entities had been clearly defined and distinguished.

Response: We thank the reviewer for pointing out this issue. By using the terms ‘circulatory’ and ‘vascular’ dysfunction, we are using the terminology set forth by the Virtual Pediatric Systems (VPS) database and do not have any further information on how these terms were distinguished. According to the VPS data set, the organ systems involved by COVID-19 identifies organ specific symptoms a patient in the database was reported to have. 

Reviewer 2

Comment: Thank you for your submission. I am curious to know why pediatric age rage was unto 21-years? It may be important to know that the studies that you have refereed/compared to have used the similar upper age. In line 15 PIM abbreviation is used for the first time without its full name kindly correct.

Response: We thank the reviewer for their comment. According to a 2017 policy statement from the American Academy of Pediatrics (AAP) entitled “Age Limit of Pediatrics,” by Amy Hardin and Jesse Hackell, the AAP acknowledges the upper age limit of pediatrics to be 21 years of age. In accordance to this statement, the US Department of Health and the Food and Drug Administration (FDA) reference approximate age ranges for late adolescence to be 18-21 years old. One very similar study we have referenced and cited as reference 38 in our manuscript entitled “Acute Kidney Injury in Critically Ill children and young adults with suspected SARS-CoV2 infection” by authors Basu et al., utilized an even broader age range of >1 week and <25 years of age. Ultimately, we chose the age of 21 due to the AAP, US Department of Health, and FDA guidelines. 

Additionally, we thank the reviewer for pointing out the oversight in line 15. The abbreviation has been defined in the manuscript.

---

## [Decision Letter · Decision Letter 1]

28 Mar 2022

Acute Kidney Injury in COVID-19 Pediatric Patients in North America: Analysis of the Virtual Pediatric Systems Data

PONE-D-21-39403R1

Dear Dr. Raina,

We’re pleased to inform you that your manuscript has been judged scientifically suitable for publication and will be formally accepted for publication once it meets all outstanding technical requirements.

Kind regards,

Zhanjun Jia

Academic Editor

PLOS ONE

Additional Editor Comments (optional):

Reviewers' comments:

Reviewer's Responses to Questions

**Comments to the Author**

1. If the authors have adequately addressed your comments raised in a previous round of review and you feel that this manuscript is now acceptable for publication, you may indicate that here to bypass the “Comments to the Author” section, enter your conflict of interest statement in the “Confidential to Editor” section, and submit your "Accept" recommendation.

Reviewer #1: All comments have been addressed

Reviewer #2: All comments have been addressed

2. Is the manuscript technically sound, and do the data support the conclusions?

Reviewer #1: Yes

Reviewer #2: (No Response)

3. Has the statistical analysis been performed appropriately and rigorously? 

Reviewer #1: Yes

Reviewer #2: (No Response)

4. Have the authors made all data underlying the findings in their manuscript fully available?

Reviewer #1: Yes

Reviewer #2: (No Response)

5. Is the manuscript presented in an intelligible fashion and written in standard English?

Reviewer #1: Yes

Reviewer #2: (No Response)

6. Review Comments to the Author

Reviewer #1: Dear Authors,

Thank you for your responses to my comments.

They have adequately addressed all that had been raised by me.

Kind regards.

Reviewer #2: (No Response)

7. PLOS authors have the option to publish the peer review history of their article (what does this mean?). If published, this will include your full peer review and any attached files.

Reviewer #1: No

Reviewer #2: **Yes: **Aasim Ahmad

---

## [Editor Report · Acceptance letter]

4 Apr 2022

PONE-D-21-39403R1 

Acute kidney injury in COVID-19 pediatric patients in North America: analysis of the Virtual Pediatric Systems data 

Dear Dr. Raina:

I'm pleased to inform you that your manuscript has been deemed suitable for publication in PLOS ONE. Congratulations! Your manuscript is now with our production department. 

Kind regards, 

on behalf of

Dr. Zhanjun Jia 

Academic Editor

PLOS ONE